# Association of Diet Quality with Low Muscle Mass-Function in Korean Elderly

**DOI:** 10.3390/ijerph16152733

**Published:** 2019-07-31

**Authors:** Mikyeong Jung, Saejong Park, Hyesook Kim, Oran Kwon

**Affiliations:** 1Department of Nutrition, Hallym University Sacred Heart Hospital, Anyang-si, Gyeonggi-do 14068, Korea; 2Department of Clinical Nutrition Science, The Graduate School of Clinical Health Sciences, Ewha Womans University, Seoul 03760, Korea; 3Department of Sport Science, Korea Institute of Sport Science, Seoul 03760, Korea; 4Department of Nutritional Science and Food Management, Ewha Womans University, Seoul 03760, Korea

**Keywords:** recommended food score, diet quality, low muscle mass-function, Korean elderly

## Abstract

There is a growing body of evidence that links nutrition to muscle mass and function in the elderly, suggesting that it has an important role to play both in the prevention and management of age-related sarcopenia. Some nutrients have been studied, but less is known about the influence of overall diet quality on the loss of skeletal muscle mass and function. This study investigated the interrelationship between the recommended food score (RFS), as an indicator of overall diet quality, and muscle mass function among the Korean elderly. The sample consisted of 521 participants (263 men and 258 women), aged >65 years, who participated in the 2014–2015 National Fitness Award project. Appendicular skeletal muscle mass (ASM) was assessed by bioelectrical impedance analysis. Low muscle mass was defined as having an ASM corrected for height lower than the cutoff value established by the European Working Group on Sarcopenia in Older People. Muscle function, assessed by handgrip strength (HGS), was defined as low if it was below the 20th percentile of elderly men and women. Low muscle mass-function, defined as low muscle mass with low muscle strength (HGS), was found in 29 men (11.0%) and 22 women (8.5%). In elderly men, the low muscle mass-function group had significantly lower RFS values than the normal group after adjustments for age, body fat percentage, drinking, smoking, education, and physical activity (*p* = 0.019). However, there was no association between RFS and muscle mass-function in elderly women. Our findings suggest that better diet quality may be associated with higher muscle mass in elderly Korean men.

## 1. Introduction

South Korea was categorized as an “Aged society” in October 2018, and those who are aged over 65 years comprise almost 15% of the whole population, which is making South Korea the fastest aging country. Aging is accompanied by various physiological changes, of which muscle mass decline is one of the most prominent features [1]. The decrease in appendicular skeletal muscle mass (ASM), which is a characterized sarcopenia [2], leads to an increase in fat mass along with a reduction of physical strength and ability to perform activities of daily living, and can even cause death [3,4].

Baumgartner et al. proposed ASM divided by height squared (ASM/height^2^) as a representative muscle index [5]. The European Working Group on Sarcopenia in Older People (EWGSOP) recommended the presence of both low muscle mass and low muscle function (strength or performance) for the diagnosis of sarcopenia [2], and this is the most commonly used definition of sarcopenia. Measurements of muscle mass using dual-energy X-ray absorptiometry (DXA) or bioimpedance analysis (BIA), muscle strength by handgrip strength (HGS), and physical performance by gait speed or short physical performance battery were required for the diagnosis of sarcopenia. Among several methods to measure muscle mass in an individual, magnetic resonance imaging (MRI) and computed tomography are the gold standards. Although DXA is known to have a high correlation with MRI, muscle swelling, or intramuscular fat deposition can lead to errors in the accuracy of X-ray transmission [6,7]. BIA, which is relatively simple and quick, has proven to be less expensive than other methods and has a relatively high correlation coefficient with DXA [8,9], although one of the limitations of BIA is that the hydration status of the person makes a significant difference in the estimation of lean mass [10,11]. Recently, the BIA has been used as a new alternative in recent studies to assess muscle mass.

Sarcopenia can be influenced by factors, such as aging, disease, physical activity, and nutrition [12]. Among them, nutritional frailty refers to the disability that occurs in old age owing to the rapid, unintentional loss of lean body mass [13]. Several age-related pathologic conditions, including masticatory disability and medications, contribute to reducing food consumption in the older population [14]. There is a growing body of evidence that links nutrition to muscle mass and function in the elderly, suggesting that it has an important role to play both in the prevention and management of age-related loss of muscle mass and function. Some nutrients have been studied, but less is known about the influence of overall diet quality on age-related muscle loss. Recent review papers [15,16,17] suggest that the Mediterranean-style diet (MED) is the “slow the progression of oxidative damage associated healthy” dietary pattern most commonly used in observational studies of muscle aging and shows the most consistent evidence. The anti-inflammatory and antioxidant properties of the MED pattern have been proposed as contributing to muscle aging.

Several food-based diet quality indices exist to evaluate individual dietary intake for the promotion of health and prevention of disease. In Korea, the recommended food score (RFS) is a relatively simple method to assess dietary quality. A diet high in antioxidant nutrients, a high RFS, and a high alternate Mediterranean Diet score (MDS) have been associated with lower oxidative stress [18]. Although age-related muscle loss could be related to a low RFS among elderly people, to the best of our knowledge, no studies have investigated this association.

Therefore, in this study, we evaluated the prevalence of low muscle mass in Korean elderly men and women (aged > 65 years) by BIA, as a good alternative for assessing ASM and the prevalence of low HGS. We then investigated the association between RFS, as an indicator of overall diet quality, and low muscle mass-function among the sample participants.

## 2. Materials and Methods

### 2.1. Study Design and Participants

This study was designed as a cross-sectional study. The National Fitness Award Project is a multi-year project (2014–2017) that presents health fitness standards associated with various health risk factors (e.g., metabolic and cardiovascular health, mental health, musculoskeletal health, senility) by adolescence, adulthood, and elderly [19]. The present study was based on elderly people aged 65 years and older, who participated in the project in 2014–2015. Among a total of 1501 participants, participants with missing data on RFS (*n* = 375), covariates, such as age, percentage of body fat, drinking, smoking, education, and physical activity (*n* = 116), and having a disease status, such as osteoarthritis, osteoporosis, backache, sciatic neuralgia, herniated cervical disc, rheumatoid arthritis or osteoarthritis, renal failure, liver disease, hepatitis (type B and C), thyroid dysfunction, diabetes, stroke, and cancer (*n* = 489), were excluded. The total sample (*n* = 521) was comprised of 263 men and 258 women aged ≥65 years. The research protocol was approved by the Institutional Review Board (IRB) of the Korea Institute of Sport Science and Ewha Womans University.

### 2.2. General and Socioeconomic Characteristics and Anthropometric Measurements

All of the subjects were interviewed by trained interviewers, to obtain general information on demographic factors, socioeconomic factors, and health-related behaviors, including age, education, marital status, smoking behavior, alcohol consumption, and RFS. Education levels were categorized as follows: Middle school (not educated, graduated from primary school, graduated from middle school), high school (graduated from high school), and college (graduated from a two-year university program or graduated from a four-year university program or graduated from graduate school or higher). The marital status was classified as reporting married or living as marred. Current smoker was defined as currently smoking or ceased smoking within the past 12 months. Alcohol consumption was defined as drinking alcohol ≥1 occasion per month. Physical activity was assessed through the Korean version of the International Physical Activity Questionnaire short form [20], and the level of physical activity was quantified as metabolic equivalent task-hours per week (MET-h/week) [21].

All of the participants’ standing height was measured to the nearest 0.1 cm with a stadiometer (Seca, Seca Corporation, Columbia, MD, USA). Body mass and lean muscle mass were measured to the nearest 0.1 kg with a BIA (Inbody 720, Biospace, Seoul, Korea). In addition, body fat percentage was estimated by the BIA. Body mass index (BMI) was calculated as body mass divided by the square of the height (kg/m^2^).

### 2.3. Recommended Food Score (RFS)

Diet quality was assessed using the RFS. This tool is a simple approach based only on foods with beneficial health effects [22] and has been applied by Kant et al. [23] and McCullough et al. [24].

In this study, we used the modified version of RFS, appropriated to the Korean diet [18]. The RFS was assessed using a questionnaire related to the previous 12 months. Forty-six foods or food groups corresponding to recommended food groups were included: Grains (1), legumes (4), vegetables (17), seaweeds (2), fruits (12), fish (5), dairy products (3), nuts (1), and tea (1). The questions had structured answer choices of yes/no to whether each food item was consumed at least once a week. The daily frequency of meals was also recorded. Participants were allocated a score of 1 if they consumed the recommended food at least weekly or ate three meals daily on a regular basis. The maximum possible score was 47 points, and higher scores indicated better diet quality.

### 2.4. Low Muscle Mass and Low Muscle Function

The EWGSOP classifies low muscle mass and quality using the skeletal muscle index (SMI, ASM/height^2^), with cutoff values of 7.23 kg/m^2^ for men and 5.67 kg/m^2^ for women [13]. The same definition was applied in this study. ASM was measured by BIA (Inbody 720) and expressed as the sum of the muscle mass of the four limbs.

Muscle function was assessed by determining the HGS, which is a widely used method to evaluate muscle function. In this study, low muscle function was defined as having an HGS below the 20th percentile values (<29.0 kg in men, <18.5 kg in women) of elderly men and women. Physical fitness test battery, including grip strength, has been verified by Choi et al. [25]. HGS (kg) was measured using a hand dynamometer (GRIP-D 5101, Takei, Niigata, Japan) as follows. In an upright posture, the participants were instructed to spread their legs as wide as the shoulder width and hold the handle of the dynamometer with the second finger node. If the handle did not fit, it was adjusted with the adjusting screw appropriately. The arm was extended straight down and maintained 15° from the body. At the “start” sound, the participant was asked to squeeze the handle with maximum force for 5 s. After performing the exercise with alternating the left and right hand twice, the highest value was recorded to the nearest 0.1 kg. Between each measurement, a 60-s resting interval was allowed. Previous use of this exercise in Korea’s national physical fitness test yielded significantly consistent results, with reliability ranging from 0.62–0.93 [25]. All parameters, including HGS, were measured by a certified, professional health and fitness instructor.

### 2.5. Statistical Analyses

Participants’ characteristics were summarized using means and standard deviation for continuous variables, and counts and percentages of subjects for categorical variables. To evaluate the significance of the differences in general characteristics, according to gender, the Student’s *t*-test (for continuous variable) and the chi-square test (for categorical variables) were performed.

The covariates in this research were statistically significant in univariate analyses or were known risk factors associated with RFS or low muscle mass-function in the existing literature. Model 1 was adjusted for age [1] and percentage of body fat [1]. Model 2 was adjusted for education [26], drinking, smoking [27], and physical activity (MET-h/week) [28,29,30], in addition to the adjustments made in Model 1. The general linear model analyses after controlling for potential confounders were used to examine the association between RFS and low muscle mass-function. All statistical analyses were performed using SAS (version 9.4, SAS Institute, Cary, NC, USA). In all analyses, statistical significance was set at *p* < 0.05.

## 3. Results

The men had a mean age of 71.9 ± 4.9 years, and the women had a mean age of 71.4 ± 5.3 years (Table 1). The mean height (*p* < 0.0001), body mass (*p* < 0.0001), and lean body mass (*p* < 0.0001) of men were higher than those in women. Socioeconomic characteristics and health-related behaviors, such as education, marital status, smoking, drinking, and physical activity were significantly different between men and women. Men were more educated, lived more often with a spouse, had higher proportions of current smoker and drinker, and had less physical activity than women.

As shown in Table 2, the means of ASM (*p* < 0.0001) and HGS (*p* < 0.0001) were higher in men than women. No significant difference in the mean RFS was observed between men and women.

In men and women, the mean SMI after height adjustment were 7.4 ± 0.7 and 6.1 ± 0.6 kg/m^2^, respectively (data not shown). Low muscle mass (≤7.23 kg/m^2^ in men and ≤5.67 kg/m^2^ in women) was observed in 35.4% of men and 27.9% of women. Low muscle function (<29 kg in men and <18.5 kg in women) was observed in 19.0% of men and 19.8% of women (Table 3).

Low muscle mass-function was defined as low muscle mass (SMI) with low muscle strength (HGS), and 29 men (11.0%) and 22 women (8.5%) were classified into the low muscle mass-function group (Table 4). In men, the low muscle mass-function group had significantly lower RFS values than the normal group after adjustments for age and percentage of body fat (*p* = 0.018), and these associations remained even after further adjusting for drinking, smoking, education, and physical activity (*p* = 0.019). There were no associations between RFS and low muscle mass-function in women.

## 4. Discussion

A previous study [31] conducted in elderly participating in the 2014–2015 National Fitness Award project, reported that RFS was positively associated with HGS, a widely used method to evaluate muscle function, in elderly Korean women. Considering that sarcopenia is defined as a combination of low muscle mass and low muscle function, it is necessary to analyze muscle mass, as well as HGS, when examining the relevance of RFS in this sample population. In the present study, we found that the low muscle mass-function group had significantly lower RFS values than the normal group among elderly men, and this association was not present in elderly women. We believe that this is the first study to show that RFS may be associated with higher muscle mass-function in elderly Korean men using BIA, which has previously been proven to be a good alternative method for assessing ASM. In this study, when comparing the low muscle function group with the normal group, the RFS was marginally low in the low muscle function group (*p* = 0.059, data not shown).

Earlier research [31] and this study used a modified RFS that includes foods characteristic of the Korean diet, such as seaweed. Some recent cross-sectional studies used tools other than the RFS to determine diet quality. Nikolov et al. [32] found that higher adherence to a MED was associated with a positive effect on appendicular lean mass/BMI in elderly German women. Kelaiditi et al. [28] reported significant positive associations between the MDS and fat-free mass and leg explosive power in healthy UK women aged 18–79 years. A study conducted in Iranian elderly aged over 55 years [29] found that adherence to the MED was associated with lower odds of sarcopenia. A longitudinal study conducted in Finland [30] showed that better diet quality, as measured by higher adherence to the Baltic Sea diet and MED, might reduce the risk of sarcopenia in elderly women. Chan et al. [26] reported that a higher dietary quality index-international (DQI-I) score and higher “vegetables-fruits” dietary pattern score were associated with lower odds of prevalent sarcopenia among Chinese older men aged over 65 years. As the components of RFS are similar to the food components of these tools, the results of this study will be assumed to be similar to the studies mentioned above.

In a recent review paper, Granic et al. [17] described the potential “myoprotective” mechanisms of MED. It was hypothesized that the complex combination of nutrients within a variety of foods in the MED has “myoprotective” effects on aged muscle, and these effects may be direct or indirect. The MED’s indirect “myoprotective” mechanisms may be expressed as a reduction in risk of age-related chronic conditions (e.g., cardiovascular diseases, diabetes, polypharmacy) associated with either induction or worsening of sarcopenic symptomatology [17]. Another direct expression of the MED may also be the amelioration of several processes implicated in sarcopenia (i.e., oxidative stress, inflammation, insulin resistance, and metabolic acidosis) [17]. In summary, the MED may benefit muscle health by simultaneously affecting several processes (in addition to oxidative stress and inflammation) that occur in sarcopenia and other age-related diseases [17]. In this study, the RFS was measured based on information about the studied food items (grains, legumes, vegetables, seaweeds, fruits, fish, dairy products, nuts, and tea). Those food items contain complex carbohydrates, unsaturated fat, fiber, antioxidants (such as vitamin C, vitamin E, and carotenoids), and minerals (such as zinc, copper, iron, selenium, and magnesium) [18]. The association between the RFS and higher muscle mass function is also likely to be linked to mechanisms similar to the potential “myoprotective” mechanisms of the MED.

The present study observed that a low RFS was associated with low muscle mass-function in elderly men, but not women. Similar gender differences were demonstrated by Chan et al. [26], who found that a higher DQI-I score was associated with a lower likelihood of prevalent sarcopenia in Chinese older men, although no such association was observed in women. Conversely, Nikolov et al. [32] noted a positive association between higher adherence to a MED and appendicular lean mass/BMI in elderly German women, but no significant associations were seen in men. The reasons for this difference based on gender is unclear [33], but Chan et al. [26] suggested that exploring the lifestyle factors might explain the gender differences in the cross-sectional finding. In our sample population, smoking and drinking were among the physical activities comprising the lifestyle factors, [34], and we found a greater proportion of current smoker and drinker among the men when compared with the women. Gender differences may also be explained by the differences in the absolute values of muscle mass and function [35,36] between men and women or in the degree of decrease in muscle mass and function [35,37] with increasing age. In these instances, the differences between men and women might be related to hormonal factors [37,38]. Further research is needed to determine the exact mechanism to explain the effect of gender disparity on the association between RFS and low muscle mass-function.

The EWGSOP recommend the presence of both low muscle mass and low muscle function (strength or performance) for the diagnosis of sarcopenia [2]. Although we did not have the data on physical performance (gait speed, standard physical performance battery, and timed-up-and-go), we were able to calculate the prevalence of low muscle mass-function using only two indicators, SMI and HGS. Therefore, this is expected to be higher than the prevalence of sarcopenia, which is calculated by applying the three criteria (muscle mass, muscle strength, and physical performance) together in other studies. In the present study, the prevalence of low muscle mass-function among our participants was 11.0% for men and 8.5% for women. Compared with the prevalence of sarcopenia in the older adults living in the community aged ≥65 years using the BIA method, the prevalence of low muscle mass-function in men (11.0%) in this study was similar to the prevalence of sarcopenia in elderly men reported in Japan (11.3%) [39], and was higher than the prevalence reported in Italy (2.6%) [40] and Taiwan (3.9%) [27]. The prevalence of low muscle mass-function (8.5%) in women in this study was lower than the prevalence of sarcopenia in elderly women reported in Japan (10.7%) [39] and Taiwan (10.6%) [27], and was higher than the prevalence reported in Italy (6.7%) [40].

The limitations of our study include the following. First, we could not determine the causal relationship between RFS and low muscle mass-function because this study is a cross-sectional design. To determine the causal association between diet quality and low muscle mass-function, further longitudinal studies are necessary. Second, only the RFS was used to assess the dietary quality, and no instruments were used for evaluating the total intake or intake of nutrients. Third, the participants who join the National Fitness Award project are interested in health, so the results may not accurately represent the general population. Nevertheless, to the best of our knowledge, this is the first study to show that RFS may be associated with higher muscle mass-function in elderly Korean men, using the BIA for assessing ASM.

## 5. Conclusions

This study demonstrates that better diet quality is associated with higher muscle mass-function in elderly Korean men. Further prospective studies of larger scale or intervention studies are needed to confirm these findings.

## Figures and Tables

**Table 1 ijerph-16-02733-t001:** General characteristics of the elderly subjects.

Variable ^1^	Total ^2^ (*n* = 521)	Men (*n* = 263)	Women (*n* = 258)	*p*-Value ^3^
Age (years)	71.6 ± 5.1	71.9 ± 4.9	71.4 ± 5.3	0.271
Height (cm)	159.1 ± 8.4	165.5 ± 5.8	152.6 ± 5.0	<0.0001
Body mass (kg)	60.2 ± 8.8	64.1 ± 8.5	56.3 ± 7.3	<0.0001
BMI (kg/m^2^)	23.8 ± 2.9	23.4 ± 2.8	24.2 ± 2.9	0.002
Body fat (%)	29.5 ± 8.2	24.0 ± 5.9	35.0 ± 6.3	<0.0001
Lean body mass (kg)	22.9 ± 4.6	26.5 ± 3.1	19.2 ± 2.2	<0.0001
Education (*n*, %)				<0.0001
≤Middle school	281 (53.9)	100 (38.0)	181 (70.2)	
High school	145 (27.8)	82 (31.2)	63 (24.4)	
College	95 (18.2)	81 (30.8)	14 (5.4)	
Marital status (*n*, %)				<0.0001
Single	165 (31.7)	42 (16.0)	123 (47.7)	
Marital	365 (68.3)	221 (84.0)	135 (52.3)	
Current smoker (*n*, %)	17 (3.3)	16 (6.0)	1 (0.4)	0.0003
Current drinker (*n*, %)	237 (45.5)	178 (65.8)	64 (24.8)	<0.0001
Physical activity (METs-h/week) ^4^	6.5 ± 6.8	5.8 ± 6.6	7.1 ± 7.0	0.031

^1^ BMI, body mass index. Marital status was defined as reporting married or living as married. Current smoker was defined as current smoking or cessation of smoking within the previous 12 months, and current drinker was defined as consuming alcohol more than once a month. ^2^ Data are presented as mean ± standard deviation, or number (percentage). ^3^
*p*-Values based on the Student’s *t*-test for continuous variables, and chi-square test for categorical variables. ^4^ METs-h/week, metabolic equivalent task hours per week.

**Table 2 ijerph-16-02733-t002:** Diet quality, and muscle mass and function of the subjects.

Variable	Total ^1^ (*n* = 521)	Men (*n* = 263)	Women (*n* = 258)	*p*-Value ^2^
Diet quality				
Recommended food score	29.8 ± 8.1	29.4 ± 8.5	30.2 ± 7.6	0.290
Muscle mass				
Appendicular skeletal muscle mass (kg)	17.3 ± 3.9	20.5 ± 2.6	14.2 ± 1.9	<0.0001
Muscle function				
Hand grip strength (kg)	28.1 ± 8.1	34.3 ± 5.9	21.9 ± 4.2	<0.0001

^1^ Data are presented as mean ± standard deviation. ^2^
*p*-Values based on the Student’s *t*-test.

**Table 3 ijerph-16-02733-t003:** The cutoff values and prevalence of low muscle mass and low muscle function.

Variable	Men (*n* = 263)		Women (*n* = 258)	
	Cutoff Value	Prevalence (%)	Cutoff Value	Prevalence (%)
Low muscle mass ^1^				
Skeletal muscle mass index (kg/m^2^)	7.23	35.4	5.67	27.9
Low muscle function ^2^				
Hand grip strength (kg)	29.0	19.0	18.5	19.8

^1^ Defined based on appendicular skeletal muscle mass adjusted for height (ASM/height^2^), as proposed by the European Working Group on Sarcopenia in Older People (EWGSOP). ^2^ Defined based on the 20th percentile of handgrip strength of elderly men and women.

**Table 4 ijerph-16-02733-t004:** Recommended food score (RFS) values according to the muscle mass-function status.

Variable	Men (*n* = 263)				Women (*n* = 258)			
	*n*	RFS ^1^	*p*-Value		*n*	RFS	*p*-Value	
			Model 1 ^2^	Model 2 ^3^			Model 1	Model 2
Low muscle mass-function group ^4^	29	26.1 ± 8.4	0.018 *	0.019 *	22	28.0 ± 7.7	0.379	0.425
Normal group	234	29.8 ± 8.5			236	30.4 ± 7.5		

^1^ Data are presented as mean ± standard deviation. ^2^ Model 1: Adjusted for age and body fat (%). ^3^ Model 2: Adjusted for age, body fat (%), drinking, smoking, education, and physical activity. ^4^ Defined as low skeletal muscle mass (appendicular skeletal muscle mass divided by the height squared ≤7.23 kg/m^2^ in men and ≤5.67 kg/m^2^ in women), as proposed by the European Working Group on Sarcopenia in Older People (EWGSOP), and low hand grip strength (<29.0 kg in men, <18.5 kg in women), defined using the cutoff values for the 20th percentile values of elderly men and women, respectively. * *p* < 0.05.

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
