# Peer review of "Association of Diet Quality with Low Muscle Mass-Function in Korean Elderly"

_ijerph, 2019, doi:10.3390/ijerph16152733_

Round 1
Reviewer 1 Report
You state on line 91 that the data collection methods are published elsewhere, however I do not have free access to this paper. Recommend adding some content here on methods for data collection and use of BIA. One of the limitations of BIA is that the hydration status of the person makes a significant difference in the estimation of lean mass. I do not agree that BIA is more accurate than imaging, using the appropriate software to analyze data. BIA is cheaper.
Line 130 I do not see any validation studies for reference norms for the GRIP-D dynamometer. You are using a percentile chart from IPAG. I could not access the percentile chart to verify the cutpoints you used in the study
Using a bar graph may be helpful to better communicate your findings- Line 174
Reviewer 2 Report
-I suggest the authors to change the evaluation of sarcopenia using the European Working Group on Sarcopenia to Asian Working Group for Sarcopenia (https://www.ncbi.nlm.nih.gov/pubmed/24461239) publish in JAMDA.
-To change the conclusion of abstract: ..."higher muscle mass-function in..." to "higher muscle mass in"
Reviewer 3 Report
This is an interesting study that adds to the current body of literature. The authors are clear that this study is cross-sectional in nature. The study is presented in a clear and concise manner.
Minor comments
Line 20 I suggest changing “with muscle mass” to “and muscle mass”.
Line 21 change “me4n and” to “men and”.
Line 84 Maybe clarify what you mean by “covariates”.
Line 101, and elsewhere - In reality you are measuring body mass in kilogrammes. Body weight = body mass x acceleration due to gravity. I accept that the term body weight is commonly used to mean body mass, however in a scientific paper I would recommend you use the term body mass.
Lines 130-138 I think a few more details would be useful here to ensure that your method could easily be repeated. Was the dynamometer adjusted to account for the different length of fingers of each participant? What was the rest period between each attempt? Did the participant keep alternating between each arm i.e. left, right, left, right, left right?
Lines 147-152 I think it would be useful if you gave some more detail about how models were adjusted for confounders and/or cited a reference. Do you mean they were added to the model as covariates?
Line 162 You have undertaken a number of statistical tests so I would recommend that a correction is made to account for the possibility that some significant differences could be due to chance. E.g. Bonferroni correction or truncated product method. I don’t think that this will affect the overall outcome of the study.
Line 165 I suggest deleting “won” and clarify the currency that you are working in. It is unclear why this threshold of income has been used. Is this based on a previous study?
Line 235 Change “daily products” to “dairy products”
